# How to Use Nested Probes Coupling to Increase the Local NMR/MRI Resolution and Sensitivity for Specific Experiments

**Mihaela Lupu** [1,2,3,4] **and Joel Mispelter** [1,2,3,4,*]

1   Institut Curie, Bât. 112, Centre Universitaire, 91400 Orsay, France
2   U1196 INSERM, Bât. 112, Centre Universitaire, 91400 Orsay, France
3   UMR 9187 CNRS, Bât. 112, Centre Universitaire, 91400 Orsay, France
4   Campus d'Orsay, Université Paris-Saclay, 15 rue Georges Clémenceau 91400 Orsay, France
*   Correspondence: joel.mispelter@gmail.com

**Abstract:** In this paper, we address resonant systems intended to be used with the commercial main resonator present on all NMR or MRI instruments. The purpose of this approach is to get an improvement regarding the spatial localization and signal to noise ratio provided by an additional smaller coil. Both coils are coupled to the same sample region, and thus, are inductively coupled through their common magnetic flux. The coupling strength is characterized by the so-called mutual inductance $M$. Two practical devices are presented. Firstly, a geometrical passive decoupled resonant system ($M = 0$) allows getting a sensitive received signal from the maximized nuclear macroscopic magnetization, excited by the main resonator and detected by the smaller sniffer coil. Secondly, a strongly coupled resonant system allows us to considerably locally improve the magnetic component of the RF near field to provide an efficient nuclear spin magnetization excitation and a high received signal. For both configurations, the behavior of the coils system regarding the amplitude of $B_1$ is addressed. Finally, specific technical hints to achieve optimum energy transfer (impedance matching) are discussed, taking into account the non-ideal RF characteristics of the involved components. Examples of MRI experiments, as well as workbench evaluations and simulations support the principles exposed here.

**Keywords:** NMR; MRI; nested probes; Radio Frequency coil; magnetic coupling; mutual inductance; wireless power transfer

## 1. Introduction

Modern nuclear Magnetic Resonance Imaging (MRI) [1] has been developed since the middle of the 1980's, essentially as clinical instruments as well as research systems specially devoted to applications on small animals imaging. Material sciences are also another application of MRI.

Most of these imaging methodologies make use of the fundamental Nuclear Magnetic Resonance (NMR) [2,3]. Discovered in 1945, NMR is better known to chemists for products analysis, to biochemists for molecular structural and dynamic investigations of biological systems, to materials engineers for solid state research, and to medical investigation as tools for clinical diagnostic on a molecular level. To this end, the so-called in vivo spectroscopy has been developed since the middle of the 1970's. Electron Spin Resonance (ESR) is seldom used for this application.

An MRI/NMR instrument [4] incorporates two devices responsible for generating a magnetic field, a permanent magnet producing a static magnetic field ranging from 0.1 to 23 Tesla and a Radiofrequency Probe (RF probe) operating at the resonant frequencies of the nuclei of interest, typically from 4 MHz to 1000 MHz. The static field ($B_0$) is responsible for spin polarization, producing a macroscopic magnetization aligned parallel to $B_0$. The

RF field interacts with the spins changing the orientation of the macroscopic magnetization. The resulting Larmor precession of the magnetization around $B_0$ induces a signal in the receiving coil. The collected tiny signal (of the order of pW) is subsequently treated by appropriate receivers, then digitized and analyzed with a PC computer. High power transmitters (from mW to kW) drive the nuclear magnetization to desired states. This preparation of the nuclear spin states is done with short intense pulses of RF applied to the RF coils that are magnetically coupled to the nuclear spin bath.

The RF probe [5,6] is mostly a resonator, tuned at the Larmor precession frequency of the nuclear spin of interest. Practically, the majority of the nuclei in the Mendeleiev periodic table are suitable for NMR experiments. We choose here to illustrate RF probe designs applied to proton ($^1$H) or sodium ($^{23}$Na) NMR. The best sensitivity (best signal to noise ratio) is obtained for proton, which is the nucleus mostly used for imaging the content and dynamic of water molecules in living bodies. The signal to noise ratio obtained with the sodium MRI of living tissues is expected to be about 10,000 smaller, as with water proton MRI. However, it provides so much functional information that it is valuable to develop this imaging methodology for both clinical and preclinical investigations.

In designing appropriate probes [7], it should be kept in mind that the electromagnetic field of interest belongs to the near field regime, meaning that the electric and magnetic components of the electromagnetic wave are decoupled in certain space regions at a distance from the source smaller than a wavelength. At larger distances, the whole electromagnetic wave propagates. The corresponding energy transported by the propagating waves is lost for NMR/MRI. The probes are therefore designed to minimize these so-called radiation losses. Exceptions are sometimes encountered as with the Travelling Waves imaging procedure [8]. This technique will not be considered in this paper.

Usually, any commercial NMR/MRI instrument is supplied with a RF probe of general use. It is a resonator producing a homogeneous magnetic component (named $B_1$) of the electromagnetic field in the volume of interest (ROI). This can be a whole body resonator with a usable bore having a diameter of around 60 cm for clinical instruments, or about 12 cm for small animal research instruments. Some old instruments, especially devoted to high field (9.4 T) micro imaging are provided with a 40 mm homogeneous RF coil included in a wide bore (80 mm) vertical magnet.

The operating frequencies of these resonators are determined by the static magnetic field $B_0$ amplitude produced by the main magnet. This can be 64 MHz or 128 MHz for most clinical instruments (with a magnet producing a field of respectively 1.5 and 3 T). In this paper, we shall describe probes at 200 MHz for proton imaging on a research instrument with a magnet of 4.7 T, and at 53 MHz for sodium MRI on the same instrument. Secondly, a 9.4 T will be used to produce detailed images of a mouse eye at 400 MHz.

The purpose of the developed devices is to use a dual coils configuration. One coil is a large probe producing a homogeneous $B_1$ field for uniform excitation within a large ROI to make a whole body proton image as a first examination. The second one is used to image, with an optimized sensitivity, a much smaller ROI in order to get structural or functional information. In both cases, we shall make use of the well-known NMR property: The sensitivity is inversely proportional to the coil dimensions, generally its diameter. While it may be easily envisaged to build its own complete probe, specifically adapted to a particular sample, we will show how to benefit from the availability of the main commercial homogeneous resonators and how to design a specific probe optimizing both the localization of the desired ROI and the sensitivity by reduction of the coil diameter.

Improving the sensitivity, a simple so-called surface coil is adequate. It is a simple loop with a diameter much smaller than the main resonator providing a substantial sensitivity gain when used as a receiver coil. The surface coil can also be used to concentrate the electromagnetic energy in a small well-defined region, allowing an improvement of the efficiency of the main resonator to produce a $B_1$ field multiplied by a factor up to 10, or more. The principle of reciprocity requires that the sensitivity will be improved by the same factor.

In designing such nested probes configuration, one has to deal with the coupling between the coils, the main resonator and the small loop or surface coil. The coupling is essentially magnetic if, as expected, the capacitance between the conductors can be neglected. The so-called mutual inductance therefore plays a central role in these designs.

## 2. Magnetic Coupling

### 2.1. Definition of the Mutual Inductance

The simplest nested probes configuration (Figure 1) that will be considered here is constituted by a transverse homogeneous, whole body, resonator, either custom made or commercial, and a unique small surface coil. The latter is designed to improve the sensitivity of the experiment in a given region of interest (ROI) in the sample space.

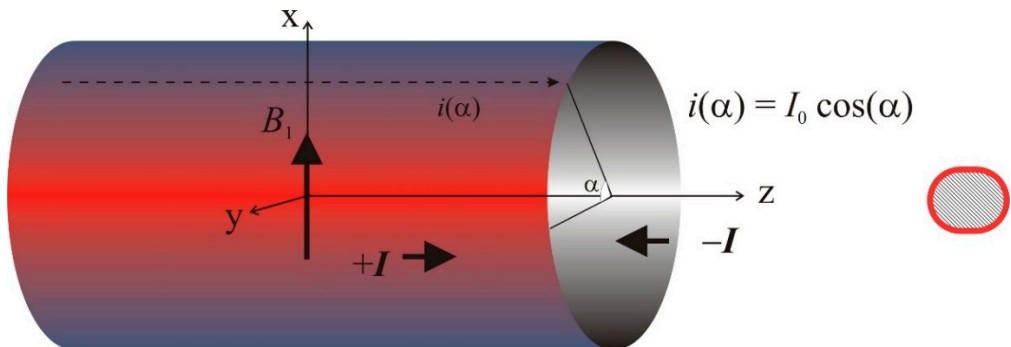

**Figure 1.** Components of the nested probe configurations described in this paper. **Left**: Schematic representation of the geometry of a cylindrical resonator and of the current density on the conductor sheet that generate a homogeneous magnetic field $B_1$ oriented along the x axis. **Right**: A typical small loop that will be inserted in the resonator.

The small surface coil is inserted at the center of the "whole body" resonator. We also assume, for the moment, that the resonator is not shielded. As will be demonstrated later, the shield, necessary in order to avoid various electromagnetic interactions with the surroundings, plays a negative role for the magnetic efficiency of the RF probe.

From the Faraday law of induction, a current is induced in the small surface coil (#2), proportional to the rate of change of the magnetic flux generated by the resonator coil (#1). The induced electromotive force (emf) from which the current is derived, is given by:

$$\varepsilon_2 = -\frac{d\phi_2}{dt}, \tag{1}$$

where $\phi_2$ is the magnetic flux generated by coil #1 through the surface of coil #2:

$$\phi_2 = \int \overrightarrow{B_1} \, ? \, d\overrightarrow{S_2}, \tag{2}$$

$\overrightarrow{B_1}$ is the magnetic induction (in Wb/m² or Tesla) created by the resonator. It is proportional to the current $I_1$ flowing in it. The integral is taken over the surface area of the closed circuit of coil #2.

Hence, the magnetic flux $\phi_2$ can be written as:

$$\phi_2 = M_{21} I_1, \tag{3}$$

where $M_{21}$ is a certain constant depending only on the geometry of the conductors.

In a similar way, a magnetic flux $\phi_1$ is induced in coil #1 by a current $I_2$ flowing in coil #2. $\phi_1$ is obviously also proportional to $I_2$:

$$\phi_1 = M_{12} I_2 , \tag{4}$$

from energy consideration [9] (p. 136), it can be shown that:

$$M_{21} = M_{12} = M , \tag{5}$$

*M* is the so-called mutual inductance. It has the same unit as a self-inductance (Henry), as it originates from the same physical phenomenon.

The mutual inductance is a measure of the magnetic coupling between both coils. If both coils shared their whole respective flux, as it would be the case in a perfect transformer, it can be shown that *M* has a maximum value equal to:

$$M = \sqrt{L_1 L_2} , \tag{6}$$

To characterize the mutual inductance in the most general case, it is useful to define a coupling coefficient *k* as:

$$k = \frac{M}{\sqrt{L_1 L_2}} . \tag{7}$$

Returning now to the coils combination, including a transverse homogeneous resonator and a small surface coil in its center, one would like to estimate the value of the mutual and the effect on the spatial distribution of the magnetic induction $B_1$. In the following, *B* will be denoted equivalently as "magnetic field" or "magnetic induction". The relevant quantity is of course the magnetic induction, expressed in Tesla unit, which is a common practice.

*2.2. Characteristics of the Transverse Homogeneous Resonator*

In all cases, the resonator generates a magnetic field $B_1$, oriented perpendicular to $B_0$, and oscillating at the resonant, operating, frequency of the nucleus of interest. The efficiency of the NMR excitation comes out from only one rotating component ($B_1^{\pm}$) of that field: the one that rotates in the same direction, and at the same rate, as the Larmor precession of the spins around $B_0$. We choose arbitrarily to display the amplitude of the component $B_1^{+}$ as the NMR efficient component. This component is derived from the decomposition of the linearly oscillating $B_1$ into two counter rotating vectors, with equal amplitudes:

$$B_1^{+} = B_1^{-} = \frac{B_1}{2} , \tag{8}$$

In some circumstances, $B_1^{+}$ and $B_1^{-}$ have different amplitudes. This has no consequences for the following discussion and will not be further considered in detail.

Nevertheless, it should be kept in mind that the total amplitude of the magnetic field $B_1$ is relevant to the magnetic coupling between coils.

Generally, depending on the size and on the frequency, the resonators may have different designs, such as the saddle coil, the Alderman Grant or slotted tube, or most frequently, a birdcage resonator.

All of these designs are approximations of the ideal configuration that generates a homogeneous magnetic field perpendicular to the static magnetic field $B_0$. The magnetic field amplitude $B_1$, at the center, of such a homogeneous transverse resonator having a diameter *d* is [6] (p. 688):

$$B_1 = K \frac{I}{d}. \tag{9}$$

In this equation, the current *I* is the total current that flows in each half cylindrical surface of the resonator. It is given by:

$$I = \sqrt{2PQ\omega C_{eq}}, \tag{10}$$

*P* is the RMS transmitter power (in W), *Q* is the quality factor of the coil, and $\omega$ is the operating pulsation ($2\pi f$, in radian per second). $C_{eq}$ (in pF) is the equivalent capacitance, related to the physical capacitance used in the device [6] (p. 688).

*K* is a factor that depends only on the geometry of the specific design of the resonator. This factor is equal to $\mu_0/2$ for the ideal, infinitely long, transverse resonator, i.e., 6.28 $10^{-7}$ T m A$^{-1}$. As a matter of fact, *K* changes only slightly among the various designs, so that an estimate of the ratio $B_1/I$ for a given transverse homogeneous resonator depends only on its diameter. Provided the length of the resonator is greater than about 1.4*d*, an average value is given by [10]:

$$\frac{B_1}{I} \approx \frac{6.8 10^{-7}}{d}. \tag{11}$$

The estimation of the mutual inductance between a given transverse homogeneous resonator and the small loop placed in its center is therefore very simple and does not require a detailed knowledge of the type of the RF main coil.

The magnetic field being homogeneous,

$$M = \frac{0.68}{d} S \cos(\theta)\ nH. \tag{12}$$

The diameter *d* of the main RF coil is in mm and the surface *S* of the small loop is in mm². $\theta$ is the angle between the direction of $B_1$ and the normal to the plane of the loop (Figure 2)

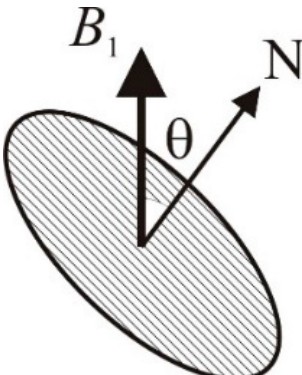

**Figure 2.** Geometry to calculate the magnetic flux through the surface of a coil as a function of the angle between the direction of $B_1$ and the normal N perpendicular to the coil plane.

As an example, consider a loop with an area of 66.3 mm² (rounded coil 10 × 8 mm) and a transverse resonator with a diameter of 40 mm, the mutual inductance is estimated to be 1.12 nH. Comparatively, the self-inductance of the loop is 20.6 nH.

It may be hard to measure such a low mutual inductance. Nevertheless, it has observable effects on the spatial distribution of the magnetic field $B_1$ within the main resonator, as well as on the electrical characteristics of the coil circuit.

### 2.3. Effects of Introducing the Small Coil into the Main Resonator

These effects are analyzed from a quasi-static simulation of the electrical and magnetic properties.

To quantitatively address these effects, one considers the following model (Figure 3). The main resonator is an unshielded, 16 leg birdcage coil. The diameter of the coil is 40 mm and the length of the legs is 70 mm. Each rung and end ring are made of 5 mm width copper strips. It is tuned to 400 MHz by 32 capacitors of 12.44 pF connecting the end rings in a high pass configuration. One of these capacitors is replaced by a parallel/series network (11.38/2.8 pF) to match the coil impedance to 50 Ohms. The Q factor of the coil is assumed to be 300, a typical value for a moderately loaded coil of this type.

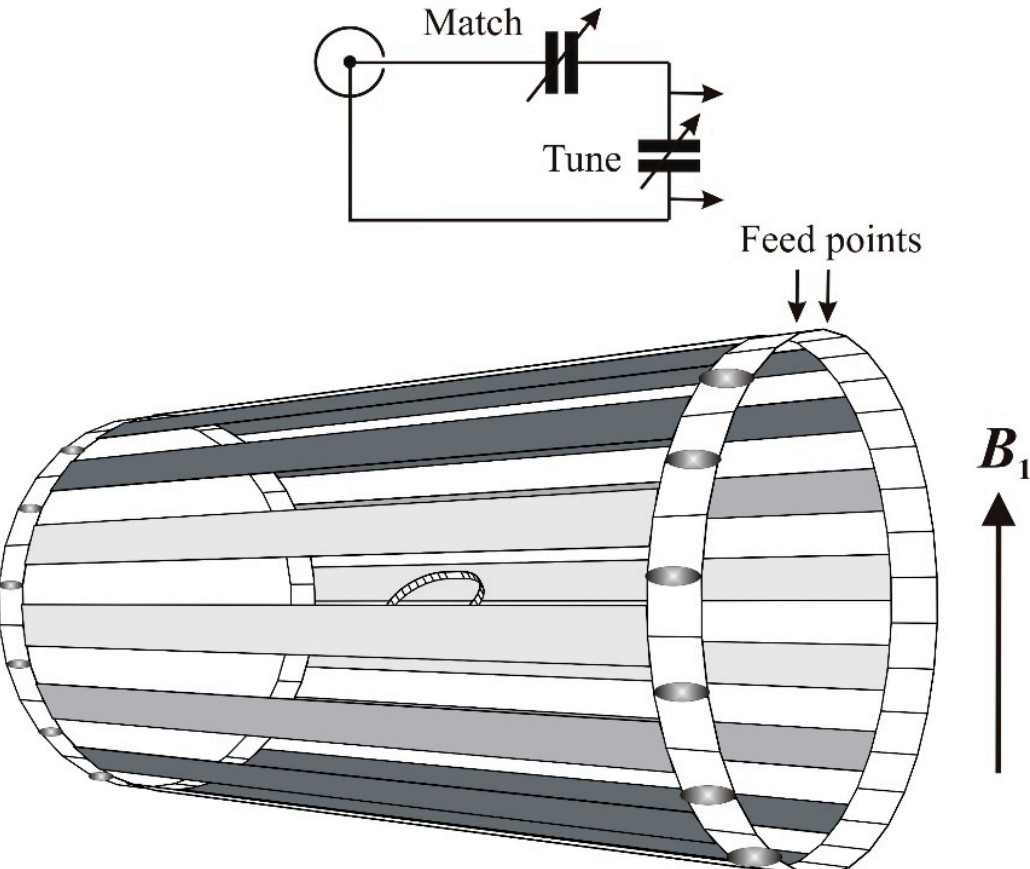

**Figure 3.** A model of the homogeneous resonator used in the simulation. The ideal homogeneous resonator is approximated by a 16 rung birdcage in a high pass configuration. The direction of $B_1$ is determined by the position of the feed points. A simple tuning/matching network is connected to the feed point as shown. A small loop is inserted in the birdcage and is visible near the center of the resonator. The dimensions and electrical characteristics are given in the text. The picture is done using the draw utility included in the FastHenry package (https://www.rle.mit.edu/cpg/codes/fasthenry-3.0-12Nov96.tar.z) (accessed on 24 January 2023).

All the electrical and magnetic simulations are made using a linear network analysis as described in [6] (p. 155–196). The NMRP software is distributed free of charge [11].

The tuning capacitances are calculated in a similar way than Birdcage Builder [12] except that all mutual inductances between the end rings are taken into account. The magnetic field generated by the probe is calculated using the Biot-Savart law, taking into consideration the contribution of currents in all pieces of the conductor.

The symmetry axis of the main resonator is aligned along z. The capacitive coupling to the transmitter is such that the resonator generates a magnetic field along its x axis. For

maximum coupling, the small loop is placed in the yz plane. For minimum (null) coupling, it is rotated in the xz plane.

The loop is, as already outlined, constituted of two half circles with centers separated by 2 mm and having a diameter of 8 mm. Its overall dimensions are therefore 10 × 8 mm. It is made up of copper strips with a width of 0.8 mm. Its Q factor is estimated to be 100, a usual value for such a coil at 400 MHz.

When the small loop, either shorted or tuned at 400 MHz with a capacitor of 7.69 pF, is placed in the xz plane ($M$ = 0) the spatial distribution of the magnetic field within the resonator is not disturbed at all, as expected (Figure 4a). The resonator remains perfectly tuned and matched at 400 MHz.

If the shorted small loop is rotated in the yz plane, its plane being perpendicular to the direction of $B_1$ generated by the main resonator, the magnetic field is clearly screened (Figure 4b). This is consistent with the Lenz law. A current is induced in the loop that opposes the generation of the magnetic flux through that coil. The amplitude of the magnetic field is reduced, in the present case, by a factor of 0.55 at the center of the coils. It is also noticeable that the resonator is slightly detuned toward higher frequency. This is consistent with the fact that the magnetic energy within the volume of the resonator is somewhat reduced, corresponding to a decrease of its equivalent inductance.

When the loop is tuned and placed in the yz plane, stronger effects are observed. The magnetic field is pumped in by the small loop. The amplification factor is about 2.4 in the present case, even if most of the power delivered by the transmitter is reflected due to a large mismatching (Figure 4c) at 400 MHz (If the circuit impedance is matched to 50 Ohms using a loss less tuner box, the multiplication factor increases to 7.5). The circuit presents two resonances as is well known [13] (p. 156) with a splitting of about 14 MHz. This splitting gives an estimation of the coupling coefficient $k$.

If $k$ is small, peaks appear at frequencies given by:

$$\nu_{1,2} = \nu_0 \left( 1 \pm k/2 \right), \tag{13}$$

$$\Delta\nu = \nu_1 - \nu_2 = k\nu_0, \tag{14}$$

The estimated coupling coefficient is thus equal to 0.035. This value agrees very well with the estimation of $k$ from Equation (7) where $L_1$ is the equivalent inductance of the birdcage coil at 400 MHz (50.9 nH), $L_2$ is the small loop inductance (20.58 nH) and $M$ is the mutual inductance already estimated as 1.12 nH. The coherence in these results gives some confidence for the validity of the simulation software.

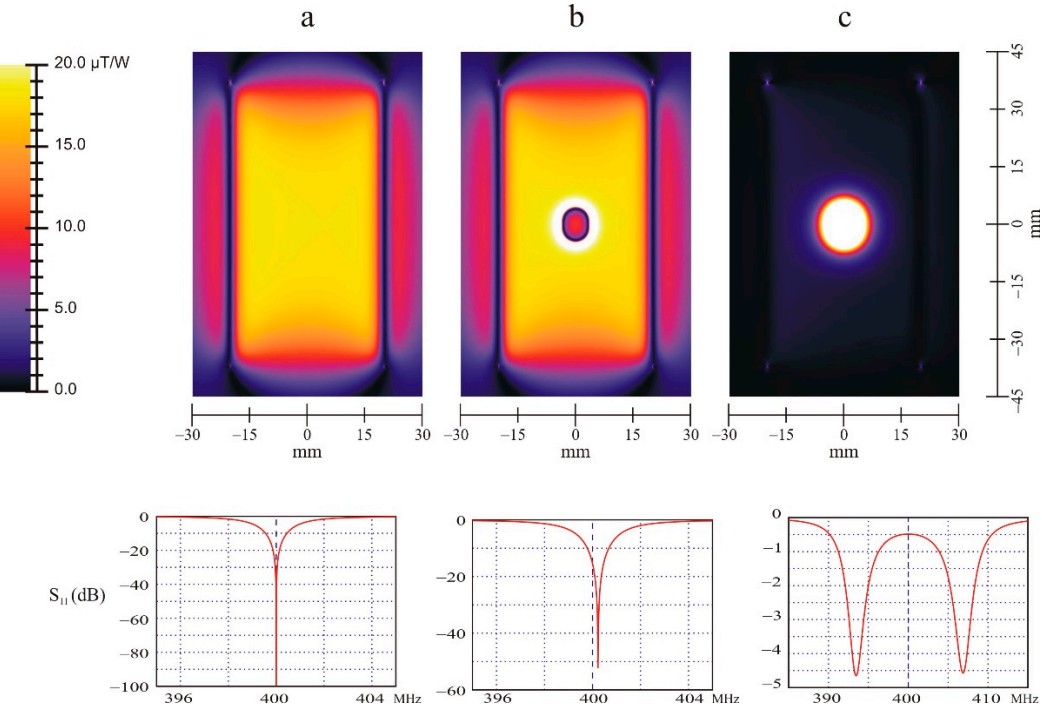

**Figure 4.** Upper: calculated magnetic field $B_1{}^+$ amplitude on the yz plane. Lower: Reflection coefficient. (**a**) The resonator alone is perfectly tuned and matched at 400 MHz. (**b**) A shorted loop is inserted in the center of the resonator. (**c**) The inserted loop is tuned at 400 MHz. The resonator tuning/matching is kept unchanged for the three cases.

The effect of inductive coupling between coils is usually avoided in a usual NMR setup. Several means are in fact proposed to decouple the coils. This can be done by active decoupling with PIN diodes activated by pulsed currents, or by so-called passive decoupling again using diodes that are activated by the high power transmitter pulses. A third way may be used for decoupling, a geometrical, passive decoupling, placing the loop in the xz plane, parallel to the direction of $B_1$. The geometrical decoupling proves to be reliable and stable in practice with the advantage of introducing a minimum of lossy noisy components, especially in the receiving channel. An example will be shown in the following.

On the other side, when the coils are maximally coupled, a large improvement of the sensitivity is expected taking advantage of the large amplification of the magnitude of $B_1$ in the region close to the tuned small loop. The advantage of this configuration is that the small surface coil is free of any electrical connection and can be integrated to the sample. No modification is required to the main resonator. It serves solely as a coupling device to the RF channel of the instrument. There are however some issues to be addressed with this configuration. The impedance at the feed point departs largely from 50 Ohms giving rise to a high reflected power during the transmit period of the NMR experiment. The obtained amplification is consequently less than it could be and the reflected power may destroy the transmitter amplifier. At best, the pulse may be shortened due to the built-in protection of the final stage of the transmitter, leading to an incomplete nuclear spin magnetization. Re-matching the main resonator is almost impossible. A remote matching is required. It will depend critically on the cable connected between the feed point of the probe and the external connector accessible to the user. This remote matching issue will be discussed here with the second example.

### 3. A Resonant System with Geometrically Passive Decoupling ($M = 0$)

The probe proposed here is designed to perform sodium imaging at 53 MHz in a horizontal 4.7 T magnet. The resonant system is used to image a small region of a mouse

having a side xenografted tumor. The sodium image of tissues is a functional imaging modality, the signal being mainly dominated by the extracellular concentration of the ions. The relaxation properties of the nuclear spin magnetization and the amplitude of the corresponding signal reflects the cellular state of the tissues [14]. This provides a diagnostic of the pathological state of any tissues and a very fast evaluation of the response to a treatment against the tumors [15–17].

The probe should also operate at the proton frequency to localize, with a good resolution, the region of interest investigated at the sodium frequency.

The probe includes a double tuned $^{23}$Na/$^1$H custom made transverse homogeneous resonator for $^{23}$Na transmitting and proton imaging and a small surface coil for receiving the $^{23}$Na signal. Due to the sodium signal being much lower than the water proton signal, particular attention should be paid to the sensitivity of the sodium channel, avoiding as much as possible unnecessary losses. The RF circuit for the sodium coil should therefore include only good quality components and only these that are essential. Magnetic decoupling of the surface coil from the resonator is however required. Only one possibility exists without inserting additional components with their associated losses, a geometrical decoupling.

The proton/sodium double tuned resonator is a birdcage coil made of 2 × 4 rungs on a cylinder of 60 mm. The birdcage is made resonant at 53 MHz in a low pass configuration and at 200 MHz in a high pass configuration. The rungs are copper strips of 7 mm width. These are tuned at 53 MHz by fixed capacitors of 90 pF and one adjustable on a rung for fine tuning. The end rings are made of trap resonant circuits having an inductive impedance at 53 MHz and an adjustable capacitive impedance to perform the tuning at 200 MHz. Finally, two opposed end rings have been removed to assure a linearly polarized magnetic field $B_1$ at both frequencies.

The birdcage feeding is made through a double tuned inductive coupling loop (90 × 35 mm) connected to a single port. The position of the coupling loop relative to the double tuned resonator is kept fixed at a distance of about 6 mm to the surface. The matching adjustment is done by two variable capacitors, acting, respectively, at 53 MHz and at 200 MHz. The latter is included in a trap circuit providing the double tune characteristics of the inductive coupling loop (Figure 5).

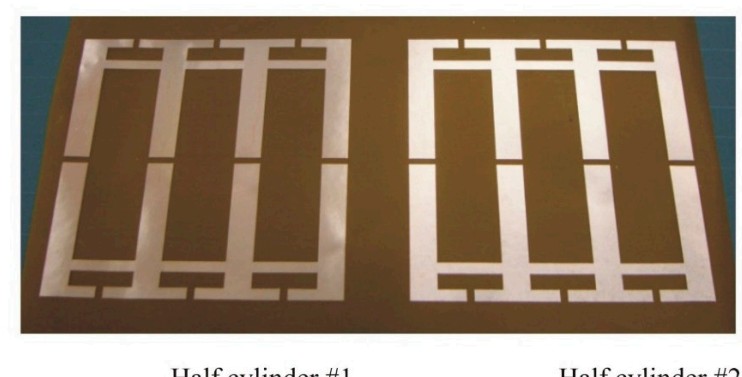

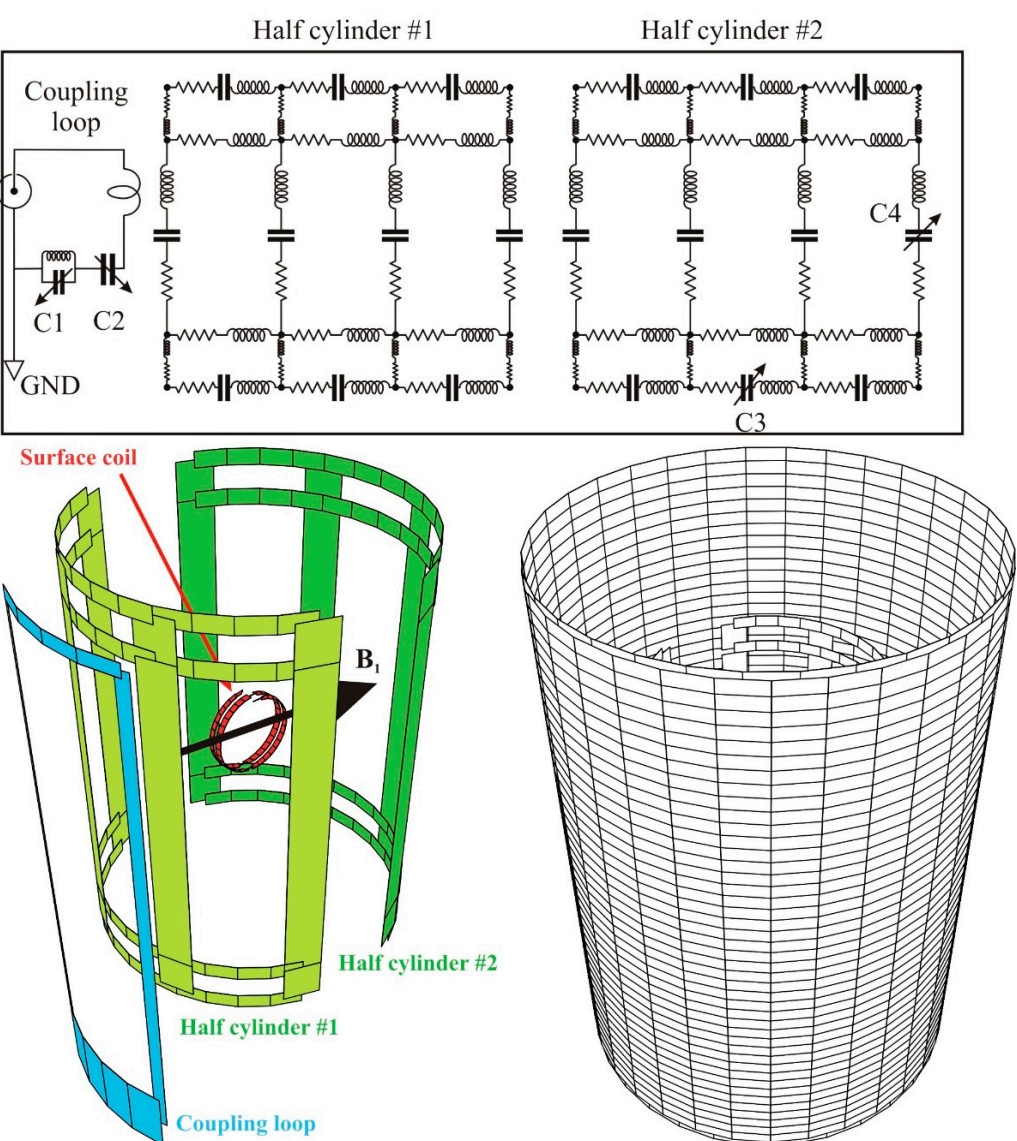

**Figure 5.** Left upper: The circuit board etched on a sheet of Kapton. The sheet is rolled up on a fiberglass epoxy FR4 cylinder with a diameter of 60 mm. The length of the principal legs ($^{23}$Na) is 60 mm, the width of the strip is 7 mm. The end rings are strips with a width of 3 mm. The traps for double tuning are constituted by loops incorporating the end rings of the sodium birdcage coil, two strips extending the sodium rungs (4.5 mm long) and another end ring (strip width of 3 mm). The tuning capacitance for sodium is equal to 90 pF. The capacitance in the proton traps is equal to 33 pF. C3 and C4 are fine tuning capacitors for respectively 200 MHz and 53 MHz. The coupling loop is also double tuned. The position of the loop with respect to the resonator is fixed. C1 and C2 match the impedance at respectively 200 MHz and 53 MHz. The trap circuit including C1 is tuned at about 235 MHz. Middle: The copper strips network of the complete birdcage coil used in the simulation. The 2 turns surface coil, tuned and matched at the sodium frequency, is represented

here in the position that permits a minimum of magnetic coupling with the resonator. Its plane is parallel to the direction of the B1 field. Right: The strip conductors network including the shield. The resonator is partly visible inside the shield cylinder (diameter of 107 mm). For the simulation, the conductors are decomposed into planar small sheets as shown here and depicted using again the draw utility of the FastHenry package [18]. Fixed capacitors are ATC nonmagnetic ceramic capacitors, package B (American Technical Ceramics Corp., KYOCERA AVX Corp. USA). Variable capacitors are nonmagnetic Voltronics trimmers (Voltronics Corporation, Knowles Capacitors, USA).

The magnetic field $B_1$ generated by this resonator is aligned perpendicular to the coupling loop at both frequencies.

Any birdcage coil has many resonant modes, at least $n$ modes for a $2n$ rungs coil. Among these modes, only one mode generates a homogenous magnetic component, relevant to the NMR/MRI experiments. With a traditional and perfectly realized birdcage coil, the homogeneous mode (m = 1) is degenerate. At the corresponding frequency, the currents in each rung located spatially at 90° from each other are equal. Additionally the RF phase of these currents are shifted by 90°. This is why the magnetic field of such a resonator is said circularly polarized if properly excited. In the present case, the degeneracy of these modes have been removed by eliminating two opposite end rings. This results in a splitting of this mode into two close frequencies. One mode generates a homogenous magnetic field when the two cylindrical halves carry current with opposed phase. The other one generates a gradient mode when the two cylinder carry in phase currents.

At a sodium frequency near 53 MHz, the homogeneous mode is expected to be at the low end of the $S_{11}$ spectrum (low pass configuration). Similarly, one expects the homogeneous mode for proton at the high end of the $S_{11}$ spectrum near 200 MHz (high pass configuration). Due to the breaking of the birdcage modes degeneracy, there are two separate resonances at each sides of the $S_{11}$ spectrum (Figure 6). The identification of the homogeneous modes is a prerequisite to conveniently tune and match the coil resonator.

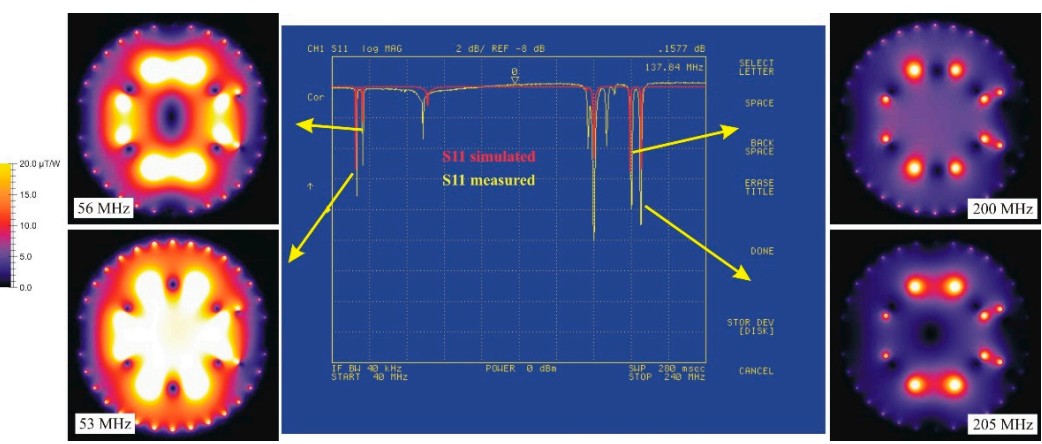

.

**Figure 6.** Identification of the homogeneous modes at 53 MHz and 200 MHz. The simulated reflection coefficient is in good agreement with the measured one, as shown in the middle. The magnetic field can then be calculated with confidence for the four frequencies corresponding to the m = 1 resonant modes of the birdcage. The homogeneous and the gradient modes are thereby well identified. One may note that the probe is not as efficient at 200 MHz as at 53 MHz. The latter frequency is obviously favored. Nevertheless, the sensitivity at the proton frequency is sufficient to localize and characterize the tissues under investigation.

The identification may be done in different ways. A simple sniffer loop placed inside the resonator identifies the homogenous modes by recording a constant signal while sampling the generated magnetic field. Another way, providing more confidence is to simulate both electric and magnetic properties of the coil using again NMRP. Figure 6 shows the measured $S_{11}$ spectrum of the designed coil. It exhibits as expected two resonant

modes, at the low and high ends of the spectrum. The simulated $S_{11}$ is in good agreement with the measured one. The simulation allows to calculate the spatial distribution of the magnetic field for the four resonant modes. It appears clearly that the homogeneous modes are assigned to the lowest frequency of each pairs of non-degenerate resonant mode. This is consistent with the general behavior of two inductively coupled tuned loops.

The resonator must be shielded [19]. The shield is segmented to minimize perturbing eddy currents that appear due to the pulsed magnetic field gradients which are required for spatial coding of the nuclei resonant frequency. This is the basis of any MRI experiment. The shield is a copper cylinder segmented along the long axis into eight large isolated strips. This provides the required high impedance at low frequency to block the DC eddy currents. On the contrary the shield should exhibit a low impedance at the high operating frequencies of the resonator. Hence, capacitors are soldered along the slits of the segmented cylinder. The shield becomes a multiple frequencies resonator. The number of capacitors and their value are chosen such that the resonant frequencies of the segmented shield are below the operating frequency range of the resonator. In the present case, the shield is a cylinder with a diameter of 107 mm and a length of 150 mm. Fifteen capacitors of 1 nF are soldered along each of the eight slits of the segmented copper cylinder.

The effect of the shield on the resonator is moderate. The ratio of the corresponding diameters of the shield and resonator is 1.78. The equivalent inductance of the birdcage coil decreases from 63.8 nH to 57.8 nH at 53 MHz. The shield has practically no effect on the inductance of the surface coil in the sodium receiving channel.

The surface coil is fixed over the animal support and positioned close to the center of the resonator. Its plane is about 10 mm above the xz plane of the resonator. The surface coil is a two turn's loop. The silver copper wire has a diameter of 1.2 mm. The coil has an internal diameter of 14 mm and a total thickness of 3 mm. Its inductance is predicted from Equation (A12) in [6] (p. 695) to be 85.7 nH.

At 200 MHz, the resonant frequency of the proton nuclei, the tuned $^{23}$Na surface coil at 53 MHz behaves as a low impedance coil, closely similar to a short circuited loop. For proton imaging, the surface coil must be therefore decoupled from the main resonator. As well, the surface coil should be decoupled from the main resonator at 53 MHz. We choose a geometrical decoupling by adjusting mechanically the relative orientation between the surface coil plane and the $B_1$ axis of the resonator.

By construction, the orientation of the $B_1$ field of the double tuned resonator at 53 MHz lies parallel to that at 200 MHz. Hence, if the surface coil plane is oriented parallel to the direction of $B_1$ (for proton or sodium), an optimum decoupling is obtained for both frequencies (nuclei).

To adjust precisely the decoupling one relies on the $S_{21}$ between the surface coil port and the resonator at 53 MHz. The $S_{21}$ has a characteristic shape shown in Figure 7 when the coupling is at a minimum. The shape is very sensitive to the relative orientation of the coil planes. It is therefore a good criteria to evaluate the quality of the decoupling. Simulations indicate that a rotation by less than 1° significantly modifies the shape of the $S_{21}$ with an associated dissymmetry of the rotating components $B_1^+$ and $B_1^-$. It is therefore important to precisely adjust the angle between the coil planes. The $S_{21}$ greatly helps for this. It proves to be highly reliable and stable.

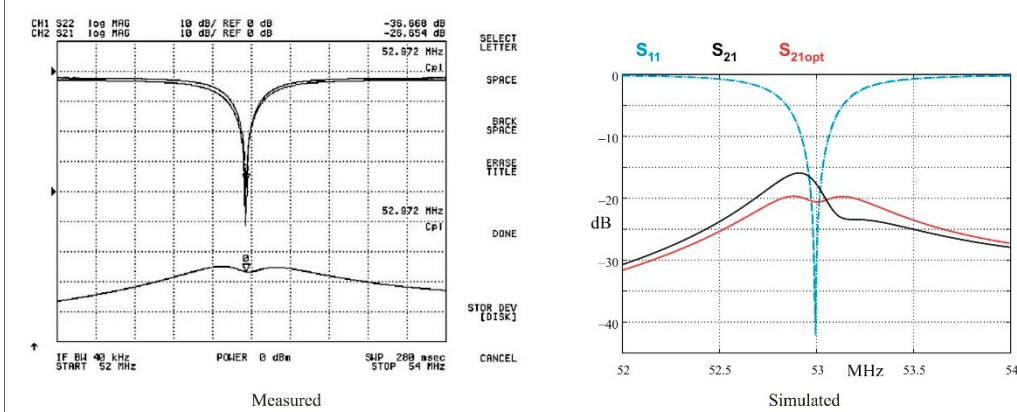

**Figure 7.** In order to adjust the geometry for the minimum of coupling between the resonator and the surface coil, one relies on the $S_{21}$ characteristic shape as measured (**left**) on a HP4396 network analyzer. The shape is well reproduced by the simulation (**right**), in red. The black trace corresponds to a slightly coupled configuration, obtained by a rotation of less than 1° of the surface coil plane from the optimum position.

If the surface coil is placed in the yz plane for maximum coupling, one observe, as described in the previous paragraph, a splitting of the resonant mode at 53 MHz. It is important to stress out that only the resonance corresponding to the homogeneous mode is split. The other resonance arising from the breaking degeneracy of the m = 1 birdcage mode is leaved invariant. This confirms the assignment of this resonance to a gradient mode with $B_1 = 0$ at center of the coil.

From the splitting (2.5 MHz), the coefficient of coupling is $k = 0.0236$. This is consistent with the coupling coefficient ($k = 0.025$) estimated from the surface coil inductance (85.7 nH), the equivalent inductance of the birdcage coil (57.8 nH) and the evaluated mutual inductance (1.74 nH) from Equation (12). This result confirms again the validity of the formulae and software simulation.

Figure 8 shows a comparison of the images and signals obtained at 53 MHz on a phantom containing a water solution of NaCl at a physiological concentration of 0.9 g per liter, in concentric tubes. The solution is diluted in agarose gels to mimic tissues environment.

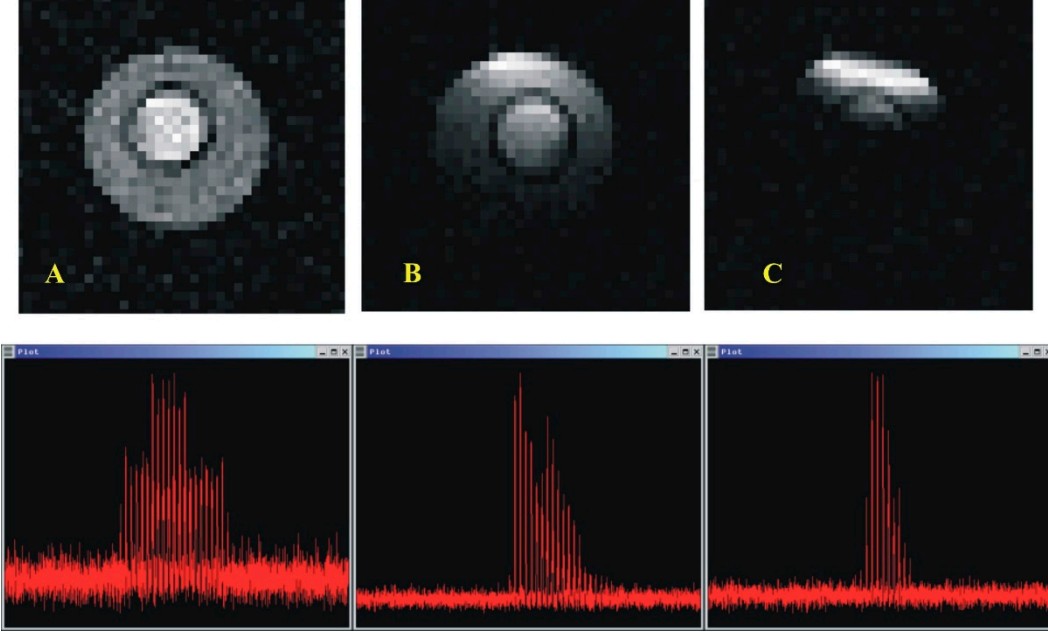

**Figure 8.** Experimental results obtained at 53 MHz on a phantom from a spin echo imaging method adapted for sodium imaging. Left (A): The image obtained with the resonator alone. The field of

view includes all the sample but with a poor signal to noise ratio Middle (B): The image obtained in a separate transmit/receive configuration is improved with respect to the field of view and the sensitivity. The benefits come out from the homogeneous excitation and from the best received signal to noise ratio provided by the surface coil.. Right (C): The image is obtained using the surface coil only. The signal to noise ratio is obviously improved, at the price of a much reduced field of view.

The first image is that obtained with the $^{23}$Na/$^{1}$H resonator in a transmit/receive mode. The trace shown below is a raster scan of the real part of the phased image. This is obtained with a phasing method developed in the lab [20]. The signal to noise ratio for the highest peak is evaluated, likewise, without any bias due to noise rectification, as 9.6.

The third image is obtained with the surface coil alone, in a transmit/receive configuration. The signal to noise increases as expected but the field of view is reduced, a characteristic of spin echo imaging with a surface coil.

The image in the middle is that obtained with the resonant system described above. The $^{23}$Na resonator, in the transmit channel, generates a uniform excitation of the nuclear magnetization. The surface coil being decoupled from the resonator has no effect at this step of the imaging experiment. The signal is then collected by the surface coil. This results in larger field of view and the best signal to noise ratio (47).

## 4. Geometrically Over-Coupled Nested Probes

Observing a particular small region of interest of living tissues in an imaging experiments or any other NMR analysis of small solid or liquid samples requires a specific probe design optimized for a maximum sensitivity. An example has been described in the previous paragraph, using separated probe channels. One is for transmitting and the other for receiving. The receiver coil is designed with a diameter compatible to the desired ROI, frequently much smaller than the diameter of the transmitting coil. This design takes advantage of the general rule that the amplitude of $B_1$ generated by a unit current is inversely proportional to the coil diameter. From the Principle of Reciprocity [21], the received signal amplitude is also inversely proportional to the coil diameter.

The device described here takes advantage of the same rule. The small coil generates locally a high $B_1$ amplitude and receives a signal with an amplitude proportional to the inverse of its diameter. Usually, this is done using a specifically designed surface coil probe connected in a transmit/receive configuration. This approach was in fact developed in the early days of ex vivo and in vivo spectroscopy [22–26], even before MRI existed.

An alternative configuration has been proposed as implanted, non-connected, coils [27,28] with several advantages regarding the simplicity of use and the reproducibility of the experimental set up for a long time follow up. A resonator, connected to the NMR instrument in a transmit/receive configuration is inductively coupled to the implanted coil. Its role is to transfer the RF energy to and from the implanted coil. The resonator is usually an already installed probe in the instrument, designed for general use. The small coil is inserted into the main resonator. The transfer of RF energy between both coils relies on the mutual inductance. The main objective is to transfer the maximum of energy into the small coil in order to get, locally, an improvement of the excitation efficiency of the nuclear magnetization and consequently of the received signal sensitivity.

### 4.1. Resonant Modes of Tuned Inductively Coupled Coils

It has been shown previously in paragraph 2.3 that, if a coil, already tuned to a given frequency $\nu_0$, is inserted inside another coil, tuned and matched to the same frequency, the resonance of the main coil is split into two new resonances.

The splitting is resolved when the coupling coefficient $k$ (Equation (7)) between both coils verifies:

$$k \geq k_c \sqrt{\frac{1}{2}\left(\frac{Q_1}{Q_2} + \frac{Q_2}{Q_1}\right)}, \tag{15}$$

where $k_c$ is the critical coupling given by [13] (p. 156):

$$k_c = 1 \Big/ \sqrt{Q_1 Q_2}, \tag{16}$$

and $Q_1$ and $Q_2$ are the quality factors of the respective coils.

As a matter of fact, resonators for NMR/MRI at high fields have typically $Q$ factors ranging from 100 to 300. Thus, the critical coupling $k_c$ ranges from 0.01 to 0.003.

The coupling coefficient depends on the orientation of the inserted coil plane respective to the direction of $B_1$ field generated by the hosting resonator (Equation (12)). When the plane of the nested coil is perpendicular to the $B_1$ generated by the main resonator, $k$ has been proved to be about 0.02 or greater for devices similar to the one already described.

In the case of over-coupled resonators tuned at the same frequency $v_0$ (present case), the currents in each coil at the lower and higher frequency modes of the splitting are respectively in phase or opposed [6] (p. 409).

At the low frequency resonant mode, the magnetic fields $B_1$ generated by each coil add together to produce an amplified magnetic field component.

At the high frequency resonant mode, the generated $B_1$ subtract together. The amplitude of the magnetic field generated by the small coil is expected to be larger than that of the large resonator. Thus, one still expects a magnetic field amplification in the proximity of the smaller coil.

When the coils are already tuned to the operating frequency $v_0$, defined by the magnetic field value $B_0$, the over-coupled coils system cannot be used as it is due to a large mismatch. The usual solution is to shift the coil resonances to a higher or lower frequency in such a way that one of the split resonant modes coincides with $v_0$. This implies a large modification of the main resonator settings, not always possible. However, recovering the state of the instrument for routine use may take a long time.

Another way is to let the main resonator without any modification. The issue is the large mismatch at the operating frequency precluding an efficient transfer of energy to the coils and potentially capable to damage the transmitter. An obvious solution to correct the mismatch is to add an appropriate matching box at the connecting port of the main resonator, but at the price of possible excessive energy losses in a critical part of the main resonator design.

This solution is nevertheless very easy to implement (Figure 9). The main resonator is firstly tuned and matched in the usual way with a representative sample of the desired experiment. The small coil, already installed on the sample, is tuned on the workbench at the operating frequency $v_0$. A small sniffer loop connected to a network analyzer can be used for this task. Then the sample with the implanted loop is installed into the main resonator. It is rotated around the z axis while observing the reflection coefficient at the resonator port to maximize the resonances splitting. Finally, a tuner box is connected to the resonator port without changing anything on the previous adjustment of the main resonator.

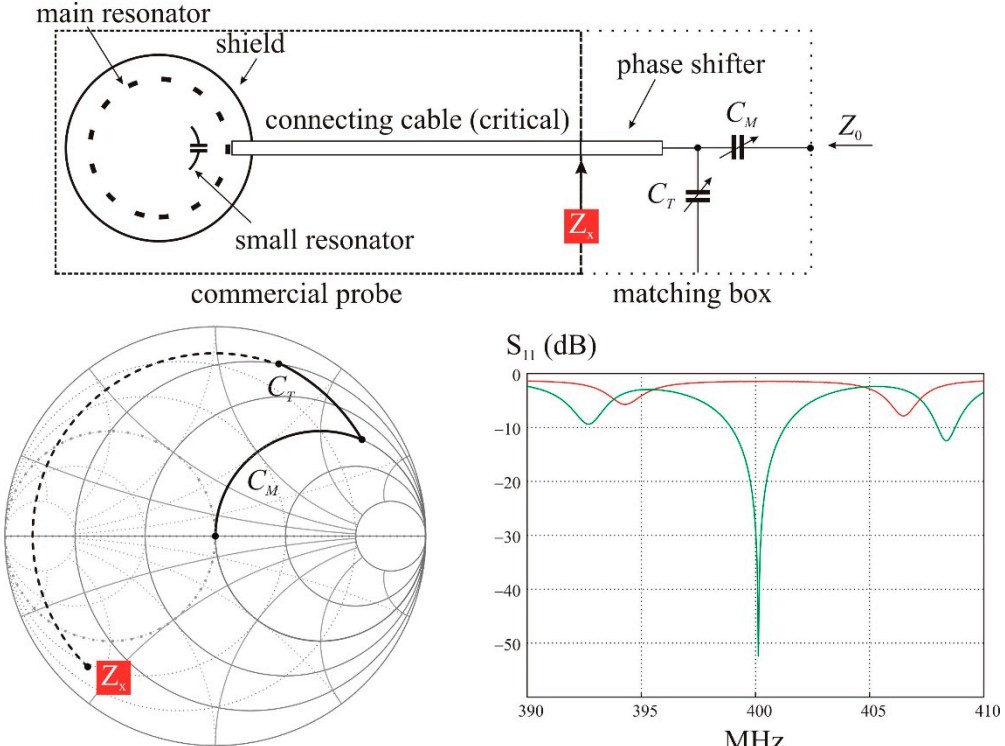

**Figure 9. Upper**: design of the magnetic flux concentrator. A small loop tuned at the operating frequency is inserted inside a volume commercial probe. The settings of this probe are kept unchanged. **Lower left**: The use of Smith chart allows solving the matching network. A short piece of line shifts the phase of the reflection coefficient (dotted line) to put the impedance in a region where a parallel/series capacitive matching network can be used. **Lower right**: The maximum coupling between the resonators is obtained when the splitting is maximum (red trace, the matching box is non-connected). Introducing the matching box into the circuit allows to obtain the required match at the working frequency (green trace).

Simulations has been performed on a model of this setup in order to investigate quantitatively the amplification factor as a function of the tuned frequency of the small coil. The simulation takes account the characteristics of the cable connecting the electrical circuit of the resonator probe to the accessible external connector. This cable proves to be the critical part for an efficient transfer of energy from the transmitter to the coils system.

### 4.2. A Practical Resonant System with Over-Coupled Nested Coils (k > 0.02)

The MRI instrument was built around a vertical wide bore (80 mm) 9.4 Tesla magnet (400 MHz for proton resonance). The gradient coils let an accessible free bore of 54 mm. The commercial resonator installed by the supplier was a Varian (The Varian NMR/MRI division was sold in 2010 to Agilent, originally Hewlett Packard. In 2014 Agilent splits into two companies, Agilent and Keysight Technology. At that time Agilent stopped the NMR/MRI activity. The corresponding pioneering activity of Varian in the domain of Magnetic Resonance unfortunately no longer exists [29]) millipede linear birdcage coil. The console was a Varian Direct Drive model associated to the VnmrJ software.

The millipede resonator, designed to provide an excellent $B_1$ magnetic field homogeneity in a cylindrical volume with a diameter of 40 mm, is modeled by a birdcage coil with 16 rungs. The shield is modeled by a network of adjacent loops (88 × 64) made of appropriate strip conductors on a cylinder having a diameter of 54 mm and a length of 140 mm.

The birdcage coil is similar to that already presented in the paragraph 2 (Figure 3), except the presence of a shield. The parallel/series impedance matching network of the birdcage coil is extended by the feeding transmission line inside the installed probe. For the simulation, one considers a typical RG402, frequently encountered in the commercial

designs. This transmission line is a low-loss semi rigid cable made of a copper tube with a diameter of 3.6 mm as a braid and a polyfluoroethylene (PTFE) dielectric. Its matched loss is 0.25 dB/m at 400 MHz, but the loss may increase when the line is not terminated by its characteristic impedance (Appendix B). In the present case, the expected loss is 0.8 dB when the line length is 0.5 m.

The line is terminated by a matching box assumed, in the simulated circuit, to be a general lossless L-matching network. In practice, to prevent the use of inductors, the matching box includes also a small length of good quality cable as the RG203. This piece of transmission line changes the phase of the transmitted and reflected waves in order to put the $S_{11}$ ratio in a region where the corresponding impedance can be finally matched to 50 $\Omega$ with a classical and easy to build parallel/series capacitive network (Figure 9).

A small loop of 8 × 10 mm is shaped on a 16 mm diameter cylinder to fit well with a mouse eye. The loop is tuned on the workbench at 400 MHz with the appropriate sample in place. It is finally inserted in the resonator and rotated around the z axis to obtain the maximum of coupling as displayed by the largest splitting in the $S_{11}$ spectrum. The loop is modeled in a plane as presented in paragraph 2, then projected on a cylinder with a diameter of 16 mm.

*4.3. Results of the Simulation and Experimental Imaging on a Mouse Eye*

The very close proximity between the shield and the resonator reduced significantly the $B_1$ field amplitude inside the sample space by a factor of $\left(d/d_s\right)^2$ where $d$ and $d_s$ are, respectively, the diameter of the resonator and of the shield [6] (p. 526). This reduces the receiving sensitivity by the same order of magnitude. The principal reason of the decrease of $B_1$ amplitude in the center of the resonator is that a large part of the magnetic energy of the RF near field, generated by the currents flowing in the birdcage rungs, is concentrated in the space between the shield and the resonator (Figure 10a).

Figure 10b shows the pumping effect already described in paragraph 2. But in addition, the magnetic energy located between the resonator and the shield is considerably reduced as well as the $B_1$ amplitude inside the resonator itself. As a result, the $B_1$ amplitude in the volume of the small coil is increased by a factor of about one order of magnitude as compared to that generated by the resonator without the loop.

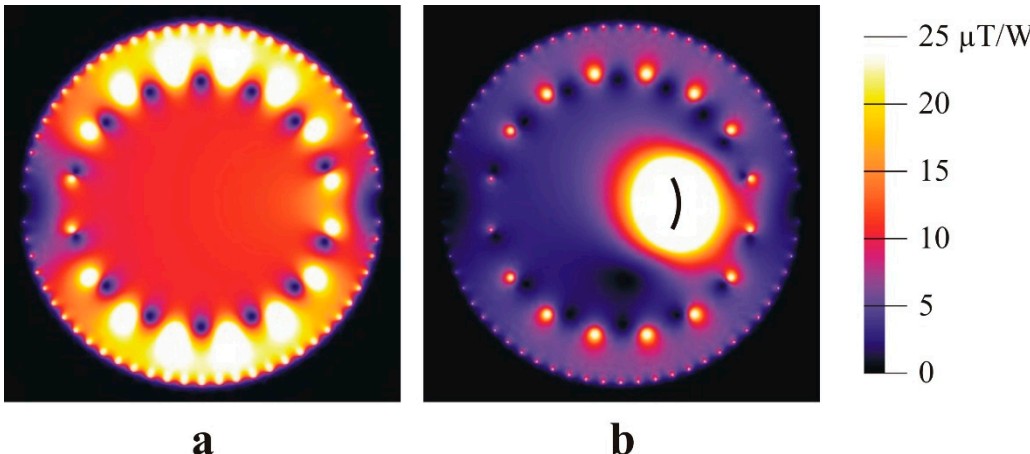

**a**                    **b**

**Figure 10.** Magnetic field simulations (quasi-static) for a model (shielded 16 rungs birdcage) of the commercial volume probe (**a**), and for the resonant system including the flux concentrator (**b**). The same transmitter power was used in both cases. The $Q$ factors were respectively 300 for the birdcage coil and 150 for the small loop resonator. Noticeably, the current in the main RF coil diminishes, contributing to a decrease of the resistive losses in this coil. Similarly, the magnetic field decreases outside the region of interest, contributing to a decrease of the sample loss.

The amplification factor has been calculated as a function of the tuning frequency of the loop (Figure 11). For each tuning frequency of the loop, the resonator settings have been left unchanged, but the matching box has been readjusted. A lossless line has been considered for this calculation to evaluate the losses originating from the resonator itself. The magnetic field amplitude is calculated on an axis perpendicular to the loop cylindrical surface and at a distance of about 8 mm towards the center of the resonator.

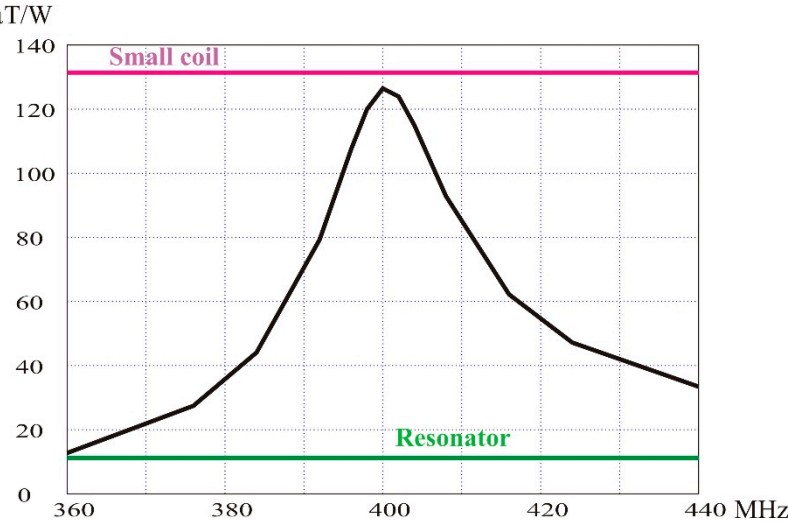

**Figure 11.** Calculation of the $B_1^+$ magnetic field amplitude as a function of the resonant frequency of the small loop. The field is calculated in the proximity of the loop surface within its field of view. The lower limit (green) is the magnetic field amplitude predicted for the shielded resonator. The upper limit (magenta) is the magnetic field amplitude generated by the small loop alone tuned and matched at 400 MHz.

As expected, the maximum of the $B_1$ field amplitude is obtained when the loop is tuned at 400 MHz. The amplitude is slightly lower than the $B_1$ field amplitude calculated at the same position when the loop is supplied by the same transmitting power. The small decrease in the magnification factor (11.3 instead of 11.7) is attributed to the losses in the resonator (0.3 dB from Equation (10)).

Finally, the loss due to the transmission line inside the resonator probe has been evaluated by simulating the circuit with a length of 0.5 m of RG402 cable connecting the matching box to the resonator circuit. The magnification factor decreases to 10.2. The difference corresponds to a total loss of 1.2 dB (from Equation (10)), of which 0.9 dB is lost in the cable. This is fully consistent with the prediction.

The amplification factor has been experimentally evaluated from a 3D image obtained on a 75 mM NaCl solution in an agarose gel to mimic the electrical characteristics of a tissue. The measured signal ratio in a slice located inside the sensitive volume of the small loop is also consistent with the prediction (Figure 12).

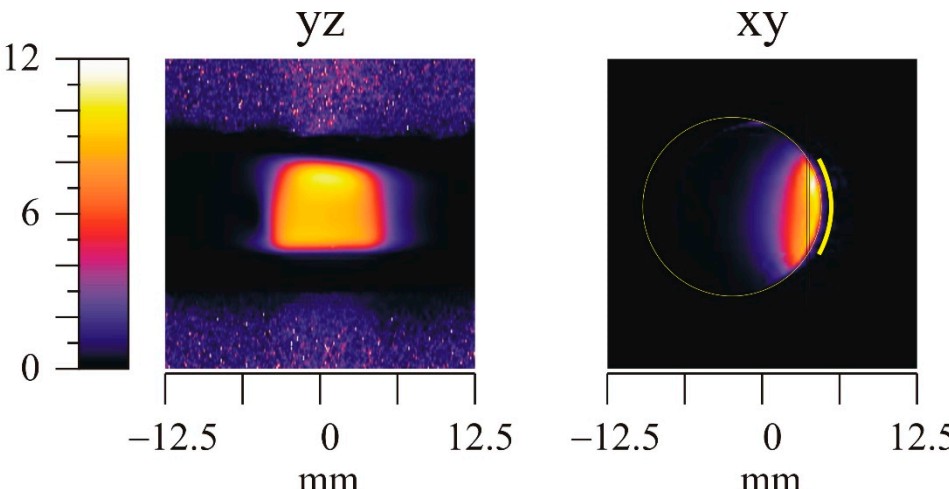

**Figure 12.** Experimental results obtained with the resonant system on an agarose sample loaded with a saline solution (75 mM). The image represents the ratio of the signal amplitudes recorded with the resonator alone and with the small coil inserted. The images are obtained with a 3D gradient echo imaging sequence. The ratio of the signal intensities is of about 10, as shown in the image on the left corresponding to the thin slice indicated on the xy orientation.

Finally, imaging on the mouse eye, an organ with a diameter of 3 mm, shows that early development of tumors in the retina could be characterized with good signal to noise ratio and resolution (Figure 13).

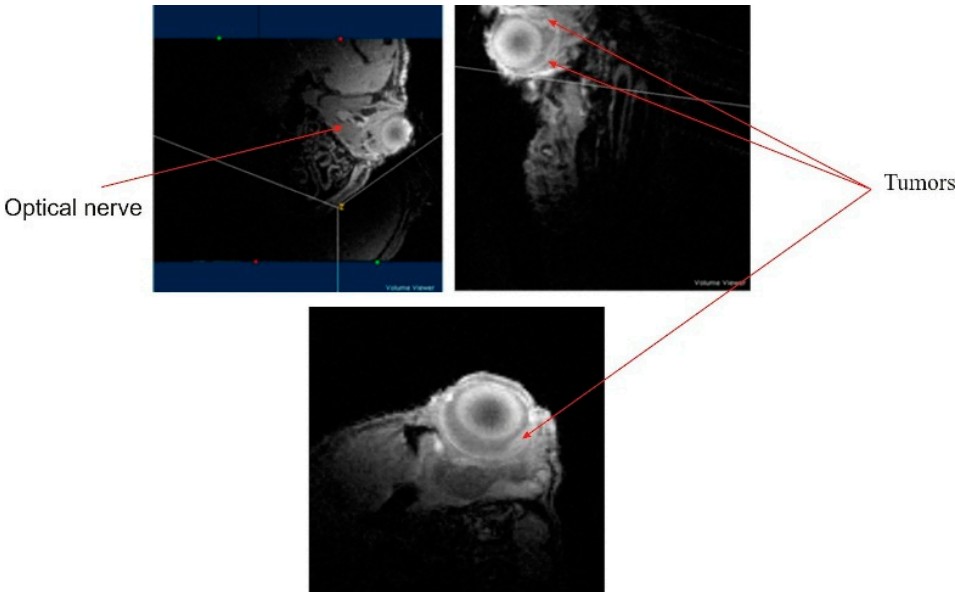

**Figure 13.** Results obtained with the resonant system presented in this work. An echo-gradient 3D image with the resolution of 100 $\mu^3$ (FOV 25 × 25 × 12.5 mm, data matrix 256 × 256 × 128) was acquired in about 40 min. The tumors developing in the retina layer are easily observable without contrast agents.

## 5. Conclusions

The first case imposes to the probe design a nearly zero mutual coupling and describes the experimental set-up to reach this condition. The geometrical passive decoupling avoids insertion of lossy components in the electrical circuit. This is very convenient when looking for a gain in sensitivity, particularly when imaging hetero-nuclei, like sodium.

The last case of resonant system uses instead the maximum mutual coupling between the nested probes in order to amplify, by an order of magnitude, the $B_1$ field in a certain region of interest. The setup is very simple as it uses an unmodified large resonator and a

single loop fixed on each sample to be examined. The loop is tuned by one capacitor at the operating frequency. Simulations, supported by the experimental results, showed that the reduction in sensitivity due to the shield is largely compensated and that the energy deposited in the whole body of the animal under examination is largely reduced. The critical part of the setup is the line relating the probe head to the external connector. This line should be of high quality. This is generally the case as encountered in a commercial probe, but should be carefully tested.

**Author Contributions:** All authors (M.L. and J.M.) contributed to the conceptualization, methodology, software, validation, investigation, writing—original draft preparation, writing—review and editing. All authors have read and agreed to the published version of the manuscript.

**Funding:** This research received no external funding

**Acknowledgments:** The authors acknowledge ARC (Association pour la Recherche contre le Cancer), PIC (Programme Incitatif et Coopératif Rétinoblastome) of Institut Curie, the French "Institut National du Cancer", the "Cancéropole Ile de France" and the French Organization "Rétinostop".

**Conflicts of Interest:** The authors declare no conflict of interest.

## Appendix A

The evaluation of the electrical properties of any probe is a prerequisite to characterize the flow of RF power within the circuit in which the probe is embedded and to estimate the perturbation of these characteristics by any interaction with the surrounding. To do this evaluation, the so-called Scattering matrix $S$ is an invaluable tool [30]. The measurement of the $S$ matrix elements is easy and permits for example (paragraph 2, Figure 4) to evaluate the perturbations generated by the insertion of a secondary loop, due to the mutual inductance $M$.

The concept of scattering matrix has been introduced for linear network analysis after the World War II [Carlin, 1954, Kurokawa, 1965, Hewlett-Packard Journal, 1967]. This concept is essential for the characterization of flow of electromagnetic energy in Radio Frequency (RF) and Microwave multi-ports circuits (filters, duplexers, RF NMR probes, active circuits, amplifiers). Shortly, when a source having a defined internal impedance is connected to a load, the optimum transfer of energy is realized when the load presents at its input an impedance equal to the complex conjugate of the source impedance.

$$Z_{load} = Z_{source}^*, \tag{A1}$$

where the asterisk denotes the complex conjugate. When this condition is realized, the available power of the generator is entirely dissipated in the load. If the input impedance $Z_{load}$ differs from the matched condition in Equation (A1), not all the available power is dissipated in the load. As a consequence, a certain amount of the power is reflected toward the source.

Any NMR/MRI Radio Frequency (RF) circuit, in which the probes are embedded, includes one or more channels designed to perform the required NMR experiments at the resonant, operating, frequency of the nuclei of interest. Each channel comprises a generator and a receiver. The generator is a high power transmitter that puts the nuclear spin magnetization in a desired non equilibrium state. The receiver collects the voltage induced by the precession of the nuclear spins around the static magnetic field $B_0$ during the subsequent evolution of the spin magnetization.

The exchange of power between the probe and the RF equipment is realized using low-loss transmission lines having a well-defined characteristic impedance. This impedance is usually real valued and independent on frequency over a very broad band. Typically the impedance is equal to 50 Ω. It determines the system impedance for every part of the RF components, except may be the input impedance of the receiver preamplifier

(LNA) which is optimized for best signal to noise ratio rather than for optimum power transfer.

Due to the choice of a system impedance with transmission line having the impedance $Z_c$ equal to 50 Ω, the general Equation (A1) becomes

$$Z_{gen} = Z_c = Z_{load} = 50\Omega \, . \tag{A2}$$

If the condition in Equation (A1) or (A2) is not fulfilled, the ratio of the reflected to the incident power is given by the square of the magnitude of a complex valued parameter:

$$\left| S_{11} \right|^2 = \frac{reflected \ \ power}{incident \ \ power} , \tag{A3}$$

$S_{11}$ is the diagonal element of the Scattering matrix representing here a one port network. If the network has two ports such as a probe with separated transmit and receive channels, the S-matrix is bi-dimensional, instead.

It is worth mentioning that the S-parameters, the elements of the S-matrix, are defined when the network is placed in particular conditions. Specifically, for a two ports network, $S_{11}$ is defined when the port 2 is terminated by a matched load and reciprocally.

The square of the magnitude of a non-diagonal element of the *S* matrix represents the energy leakage from one port to another. For a two-ports network:

$$\left| S_{21} \right|^2 = \left| S_{12} \right|^2 = \frac{transmitted \ power \ at \ port \ 2 \ or \ 1}{incident \ power \ at \ port \ 1 \ or \ 2} . \tag{A4}$$

The magnitude of a diagonal element $S_{ii}$ varies from 0 (perfect match) to 1 if the load is either opened or shortened. It is frequently expressed in dB in which case it represents directly the reflected to incident power ratio. The off-diagonal element $S_{ij}$ varies also from 0 to 1. $S_{ij}$ = 1 corresponds to a transmission between ports *i* and *j* without any loss, while 0 corresponds to the case when there is no coupling between ports.

The phase of $S_{ii}$ represents the reactive component of the load impedance. The phase is positive for an inductive load and negative for a capacitive load. It is 0° for an open load and 180° for a short. The phase of $S_{ij}$ denotes the change of phase of the voltage associated with the outgoing wave at port *i* respective to that of the incident wave at port *j*.

A network analyzer incorporates a frequency sweeper and one or more receivers to measure the voltages, instead of the power (This is permitted as far as the impedances are all referred to 50 Ω), associated with the incident and reflected wave separated by an appropriate device (bridges or directional couplers). Depending on the complexity (and the price) of the network analyzer, one can get either the magnitude of *S* using a Scalar Network Analyzer (SNA) or the real and imaginary component of the complex valued *S* using a Vector Network Analyzer (VNA) [30]. The latter are quite expensive instruments. The models designed for the frequency range of NMR/MRI instruments cost from ten to forty thousand USD, or more. Recently, some very cheap instruments have been proposed on the market, like the nanoVNA. These are capable of reasonably accurate measurements for bench-work evaluation of NMR probe at a price between 50 to 150 USD. More, they allow the final matching/tuning adjustments at the proximity of the magnet.

**Appendix B**

The critical part in the probe circuit is the transmission line relating the resonator to the external connector of the impedance matching box. This line being terminated by an impedance differing from the cable characteristic impedance, larger losses than expected from the datasheets are possible.

The ratio of the power delivered into the load to the power applied at the input of the cable is given by [6] (p. 74):

$$\frac{P}{P_{in}} = \frac{\left(1 - |\Gamma|^2\right)}{\left(A - |\Gamma|^2 / A\right)},$$
(A5)

where $A$ is the attenuation factor of the line terminated by its characteristic impedance and $|\Gamma|$ is the magnitude of the reflection coefficient at the load.

Considering a low-loss line, the matched loss is:

$$ML_{dB} = 10 \log_{10}(A)$$
$$A = e^{2\alpha l},$$
(A6)

$l$ is the length in meter and $\alpha$ is a factor (in Neper/m) depending on frequency as:

$$\alpha = a\sqrt{f} + bf,$$
(A7)

For a RG402 cable, $a$ and $b$ are respectively equal to 0.012 dB/m and $3.10^{-5}$ dB/m when $f$ is in MHz. At 400 MHz, the matched loss in such a line 1 m long is 0.252 dB, but if the reflection coefficient approaches 1, the loss in the line increases to 9 dB (Figure A1).

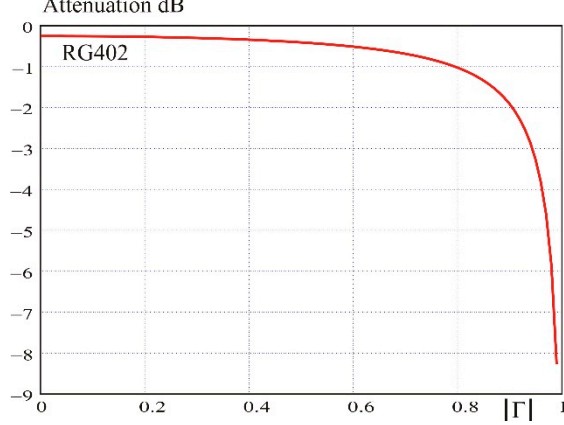

**Figure A1.** Attenuation of a 1 m of RG402 semi rigid coaxial cable, at 400 MHz, as a function of the reflection coefficient at the load terminating the line. If the line is terminated in its characteristic impedance ($|\Gamma| = 0$), the matched loss is 0.25 dB. The attenuation increases smoothly to 1 dB at $|\Gamma| = 0.8$, then very quickly above this reflection coefficient.

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
