# Peer review of "How to Use Nested Probes Coupling to Increase the Local NMR/MRI Resolution and Sensitivity for Specific Experiments"

_electronics, doi:10.3390/electronics12030594_

Round 1

Reviewer 1 Report

In this paper, the authors describe a method with which to enhance the local sensitivity of the NMR/MRI probe by inserting another coil, specifically optimized to achieve high sensitivity for the region of interest within the sample, in the NMR/MRI primary probe. The authors described two scenarios of coupling the introduced coil to the primary one. They used simulations and experimental studies to demonstrate the sensitivity enhancement due to the proposed method. I find this paper a nice contribution and would therefore recommend accepting it after minor revisions. Here are some points that need to be revised:

- There are several grammar and phrasing mistakes, for example:

line 16: allows getting

line 22: taking into account

line 147: replace "than" with "as" 

line 267: built-in

line 286: as much as 

line 438: issues => tissues

I recommend an extensive review of the English grammar and phrasing.

- line 43: that is not completely true! Conventionally, the static field is responsible for spin polarization whereas the RF field is responsible for spin manipulation. 

- equation (10): is not used in the text and the symbols are not explained. 

- please include the MRI imaging parameters for the reported experiments.

Author Response

In this paper, the authors describe a method with which to enhance the local sensitivity of the NMR/MRI probe by inserting another coil, specifically optimized to achieve high sensitivity for the region of interest within the sample, in the NMR/MRI primary probe. The authors described two scenarios of coupling the introduced coil to the primary one. They used simulations and experimental studies to demonstrate the sensitivity enhancement due to the proposed method. I find this paper a nice contribution and would therefore recommend accepting it after minor revisions. Here are some points that need to be revised:

- There are several grammar and phrasing mistakes, for example:

We have corrected all the errors as suggested. We thank the reviewer for these positive suggestions.

I recommend an extensive review of the English grammar and phrasing.

We have reread carefully the manuscript and we hope to have made improvements of the revised version.

- line 43: that is not completely true! Conventionally, the static field is responsible for spin polarization whereas the RF field is responsible for spin manipulation.

That's true. We have added a short description removing these inaccuracies 

- equation (10): is not used in the text and the symbols are not explained.

The parameters for this equation have been explained. It is also used after Figure 11. 

- please include the MRI imaging parameters for the reported experiments.

We have specified the imaging method (spin echoes or gradient echoes) which is important when using a specific probe configuration. The detailed parameters are however out of the scope of this paper.

Reviewer 2 Report

In this paper, the authors discuss the coupling of two types of resonant systems in MRI commercial main resonators, namely the two cases of maximum and minimum mutual inductance M, as well as the impedance matching problem of subsequent cable connections. However, it is vital important for the authors to offer the novel significance of the two practical devices design, not only for the simple examples of verification. In order to support the global remarks, the suggestions and comments are as follows.

1.In the introduction part, the necessity of the proposed structure is not fully explained, as well as what problems it solves. It is suggested to classify other types of resonance devices commonly used at present and compare them with the proposed structure. That is, the authors should explain why they wrote the article.

2.Figure 5 shows the structure designed by the authors. In this part, I hope the authors can expand the idea of structural decoupling design, how to design the parameters.

3.Part 2.1 “Definition of the mutual inductance” is some knowledge familiar to the public. It does not need to be introduced in detail in the main body. It is suggested to put the detailed content in the supplementary materials.

4.Figure 13 only shows the MRI images of the eyes in the research system designed in this paper, lacking the images of the control group. Can you compare the images before and after adding the resonant system, and compare the specific signal-to-noise ratio parameters before and after adding the resonant system? In addition, the experimental parameters in Figure 13 should be specified.

5.As a review, many pictures in this paper are not marked with references. If they are quoted pictures, please mark the references.

6.There are many grammatical errors, the author should pay attention to them in writing.

Author Response

In this paper, the authors discuss the coupling of two types of resonant systems in MRI commercial main resonators, namely the two cases of maximum and minimum mutual inductance M, as well as the impedance matching problem of subsequent cable connections. However, it is vital important for the authors to offer the novel significance of the two practical devices design, not only for the simple examples of verification. In order to support the global remarks, the suggestions and comments are as follows.

1.In the introduction part, the necessity of the proposed structure is not fully explained, as well as what problems it solves. It is suggested to classify other types of resonance devices commonly used at present and compare them with the proposed structure. That is, the authors should explain why they wrote the article.

The paper is expected to provide a global view of how one can play with the mutual inductance for designing NMR/MRI probes, optimized for a specific sample. The objective is educational. We want to share our long experience with readers, especially students and beginners in the domain of NMR probe design.

2.Figure 5 shows the structure designed by the authors. In this part, I hope the authors can expand the idea of structural decoupling design, how to design the parameters.

The question is obscure. Anyway, we have provided some details (dimensions, capacitances) in the legend of the Figure. 

3.Part 2.1 “Definition of the mutual inductance” is some knowledge familiar to the public. It does not need to be introduced in detail in the main body. It is suggested to put the detailed content in the supplementary materials.

The estimation of the order of magnitude for the mutual inductance is of importance. This part recalls the essential physical concepts related to magnetic coupling, often misunderstood and misused.

4.Figure 13 only shows the MRI images of the eyes in the research system designed in this paper, lacking the images of the control group. Can you compare the images before and after adding the resonant system, and compare the specific signal-to-noise ratio parameters before and after adding the resonant system? In addition, the experimental parameters in Figure 13 should be specified.

The gain in the signal intensity is given in Figure 12, on a phantom, and discussed in the text. The Figures 13 is shown only to give an example of what we can expect with the final design. A noisy image on which we are unable to distinguish the features i the eye retina does not provide any useful information.

5.As a review, many pictures in this paper are not marked with references. If they are quoted pictures, please mark the references.

All the pictures presented here are original and were never published. These come from the author's daily work.

6.There are many grammatical errors, the author should pay attention to them in writing.

The authors have 50 years of scientific experience and thousands pages of peer review international publications. We are used with such a comment. We have obviously reread and tentatively corrected remaining errors.

Round 2

Reviewer 2 Report

I have read the revised manuscript.Unfortunately, the revised version of this manuscript shows relatively little improvement, and for most cases, the authors engaged in a rebuttal rather than a constructive consideration.Likewise, in response to the remaining original comments the authors have provided rather vague answers, and did not include the requested supporting information / data with the revised manuscript.To this end, the conclusion is that the revision is not satisfactory and publication is not recommended.

The author did not reply directly to the questions raised. For example, as for comment 1, as a review, we hope that the author could revise the introduction part to fully discuss the research status in the field, supplement the necessity of the structure designed in the paper, and compare it with other types of structures in the field. However, the author's reply is more of a rebuttal without giving a convincing answer. In addition, as for comment 2, Figure 5 is the structure designed by the author. As an important part of the paper, it is hoped that the principle and ideas of structure design can be supplemented in the text, but the author's reply avoids this problem. Finally, as for comment 4, Figure 13 is the picture obtained by the author's own experiment to verify the effect of the structure designed by the author. In order to evaluate and understand the imaging effect of this structure, we hope the author adds experimental parameters, but it is not given in the author's reply. 

Author Response

We answered previously to the comments, according to our own opinion about this proposed paper. We regret that this does not correspond to the standards of Electronics. We accept that "the publication is not recommended", so we give up.